# Influence of Oil Viscosity on the Tribological Behavior of a Laser-Textured Ti6Al4V Alloy

**DOI:** 10.3390/ma16196615

**Published:** 2023-10-09

**Authors:** Marjetka Conradi, Aleksandra Kocijan, Bojan Podgornik

**Affiliations:** Institute of Metals and Technology, Lepi Pot 11, 1000 Ljubljana, Slovenia; aleksandra.kocijan@imt.si (A.K.); bojan.podgornik@imt.si (B.P.)

**Keywords:** oil lubrication, surface modification, Ti-based alloy, tribology

## Abstract

Laser texturing with a dimple pattern was applied to modify a Ti6Al4V alloy at the micro level, aiming to improve its friction and wear resistance in combination with oil lubrication to optimize the performance in demanding industrial environments. The tribological analysis was performed on four different dimple-textured surfaces with varying dimple size and dimple-to-dimple distance and under lubrication with three different oils, i.e., T9, VG46, and VG100, to reflect the oil viscosity’s influence on the friction/wear of the laser-textured Ti6Al4V alloy. The results show that the surfaces with the highest texture density showed the most significant COF reduction of around 10% in a low-viscosity oil (T9). However, in high-viscosity oils (VG46 and VG100), the influence of the laser texturing on the COF was less pronounced. A wear analysis revealed that the laser texturing intensified the abrasive wear, especially on surfaces with a higher texture density. For low-texturing-density surfaces, less wear was observed for low- and medium-viscosity oils (T9 and VG46). For medium-to-high-texturing densities, the high-viscosity oil (VG100) provided the best contact conditions and wear results. Overall, reduced wear, even below the non-texturing case, was observed for sample 50–200 in VG100 lubrication, indicating the combined effect of oil reservoirs and increased oil-film thickness within the dimples due to the high viscosity.

## 1. Introduction

We experience wear and friction in ordinary life when two moving components come into contact. On an industrial level, friction increases energy consumption, while wear can decrease the lifespan of components. The combination of both negative effects can significantly increase production costs and consequently lifetime expenses. Therefore, the degree of friction and wear needs to be minimized in such mechanical systems. Lubrication is a well-established approach to improve the tribological properties of components [1].

Synthetic oil lubricants play the leading role in liquid lubrication [2,3,4,5,6,7,8]. Their main function is to form an oil film between two surfaces in contact, leading to their separation and consequently reduced friction and wear [1]. Friction and wear are strongly corelated with the viscosity, type, and temperature of the lubricant as well as the surface roughness [9]. The efficiency of the oil’s penetration between the contact surfaces depends on the nature of the lubricant, the contact geometry, and the sliding conditions. Michaelis et al. [10] showed that high-viscosity lubricants can improve bearing systems, pointing out that synthetic oil lubricants have fewer power losses than mineral ones. Hu et al. [11] studied the influence of viscosity on the friction and wear properties of synthetic ester oils in terms of physical/chemical oil adsorption. Wang et al. [6] reported on the influence of the molecular structure of lubricants on friction mechanisms. Furthermore, as reported by Qin et al. [12,13], the surface roughness strongly affects the characteristics of the contact interface. The improvement of the sliding performance has been conventionally based on the optimization of the macro characteristics of the contact surfaces, i.e., machining to the desired shape or a heat treatment to enhance the surface strength [14]. Due to the increasing demands of the working conditions and severe environments it is necessary to upgrade appropriate lubrication with the use of surface-modification techniques aiming to improve the surface characteristics at the micro level. Laser texturing has already been identified as an appropriate surface-modification technique that enables tailoring of the surface properties through the manipulation of the processing parameters, leading to desirable surface topographies [15,16,17,18,19]. Different texturing geometries, such as dimples, lines, squares, cones, rhombi, and ripples, have already served as traps for wear debris, which results in a decrease in the abrasion through minimized wear–particles interactions and lubricant retention [20,21].

The majority of studies focus on the interaction between representative lubricant parameters and the tribological properties when in contact with specific surfaces. The novelty of the present study is an upgrade to the systematic analysis of the correlation between different oil viscosities and friction/wear via the surface laser texturing of a Ti6Al4V alloy. In this relation, the synergistic effect between the oil’s viscosity and the specific surface morphology was revealed. The optimum surface texture/oil viscosity was proposed to minimize the tribological loss that is crucial in demanding industrial environments. Here, the tribological analysis of a dimple-laser-textured Ti6Al4V alloy lubricated with three different synthetic oils—T9, VG46, and VG100—was performed. Even though the Ti6Al4V alloy is commonly used in automotive, aerospace, and medical applications [22,23,24,25,26] due to its high strength, fracture toughness, low density, good corrosion resistance, and biocompatibility, its major disadvantages—high friction and poor wear resistance—limit its use in harsh tribological applications [27]. To overcome these drawbacks, we have implemented laser texturing to modify the Ti6Al4V surface with a dimple-texturing pattern. We analyzed the effect of the dimple diameter and their center-to-center distances on the friction and wear in combination with lubrication based on different oil viscosities.

## 2. Materials and Methods

Materials—The material under investigation was a Grade-5 Ti-6Al-4V alloy with 6% aluminum and 4% vanadium in the solution-treated-and-aged condition (37 HRC). The samples were in the form of discs with a diameter of 25 mm and a thickness of 1.5 mm. The samples were ground with 800-grit paper. The resulting average surface roughness of the prepared discs was Sa = 0.17 µm. Prior to further use, all the discs were ultrasonically cleaned in acetone.

For lubrication, three different, commercial, synthetic base oils with different viscosities were used—Nynast T9, Nybase ISO VG46, and Nybase ISO VG100— and were all provided by Bureau Veritas (Oil & Petrochemicals). The viscosity data of the oils are listed in Table 1.

Laser texturing—Surface laser texturing was performed with a Rofin SMD 50 W II Nd-YAG laser power source and F-Theta-Ronar lens with a focal length of 160 mm. Rofin LaserCAD software was employed to program the Ti6Al4V surface’s modification with dimples having two different diameters (50 µm, 100 µm), three different center-to-center distances (100 µm, 200 µm, 400 µm), and a dimple depth of 20 µm. Laser texturing was performed in the CW mode with a pulse length of 0.2 ms, frequency of 500 Hz, pulse spacing of 0.1 mm, and a minimum pulse spacing of 0.05 mm. The 50 µm dimples were processed with 17 pulses and an electrical current of 37.0 A, and 100 µm dimples with 11 pulses and an electrical current of 41.0 A. All the texturing parameters for 4 different samples are listed in Table 2. The surface modification was performed in air at room temperature.

Surface characterization—The surface characteristics of the dimple-textured Ti6Al4V and the wear tracks were determined using a field-emission scanning electron microscope (SEM JEOL JSM-6500F). Additional surface topography and wear volume analyses were performed with a 3D optical microscope (high-resolution Alicona Infinite Focus G4) and IF-MeasureSuite (Version 5.1) software.

Tribological testing—A TRIBOtechnic friction-testing tribometer in the ball-on-flat contact configuration under a reciprocating sliding motion was employed for the tribological tests. Tribological testing was conducted under ambient conditions, with the following testing parameters: normal load of 5 N, corresponding to a nominal contact pressure of 600 MPa; average sliding speed of 5 mm/s (frequency 0.25 Hz and amplitude 10 mm); and a total sliding distance of 1 m (sliding time ~200 s). A 100Cr6 bearing-steel ball (10 mm diameter) was used as a stationary counter-body and loaded against a laser-textured Ti6Al4V disc. Sliding was performed along the dimple-texturing pattern as shown in Figure 1. Prior to testing, both contact bodies were cleaned in an ultrasound bath with ethanol and then dried in air. For lubrication, a drop of oil was applied to the surface, enabling boundary/mixed lubrication throughout the test. Each test was repeated three times, which allowed us to calculate the average steady-state coefficient of friction, COF (last 100 s). Wear tracks were analyzed with the SEM and the optical 3D microscope to determine the width and the depth of the wear scars for a calculation of the wear volumes.

## 3. Results and Discussion

Figure 2 presents the morphology of the laser-textured Ti6AL4V surface. The surface was modified with a dimple texture, with dimples sized 50 µm and 100 µm in diameter and center-to-center distances of 100 µm, 200 µm, and 400 µm. We should stress that after laser texturing, characteristic bulges were observed around the dimples, which were removed prior to the tribological testing so as not to interfere with the friction and wear. As reported in [28], the material in the bulges is harder than the base material. For this purpose, all the laser-textured samples were ground with paper down to 2400 grit. Although the hard bulges were removed, the hardness of the surface material around the dimples was still higher than the base material (i.e., 300 HV0.01 vs. 240 HV0.01).

The four laser-textured surfaces were distinguished by a comparison of the textured and non-textured areas. For this purpose, we defined the parameter texture density (TD) as the ratio between the laser-textured area (dimples) and the total surface area. The TD was determined from SEM images and is reported in Table 3. We can see that the surface 50–100 has the highest TD (around 40%), while the surface 100–400 has the lowest TD (around 10%).

To evaluate the influence of the laser texturing on microstructural–crystallographic characteristics, we performed an additional SEM analysis of the cross-section of the dimple-textured Ti6Al4V surfaces. In Figure 3, the SEM image of the dimple on the 100–400 dimple-textured surface is presented as the newly formed thin layer at the dimples, and this is similar for all four laser-textured surfaces due to the same processing parameters. The as-received material with an original microstructure, as reported by the manufacturer, was annealed at the α + β region and then cooled down to obtain polycrystalline α grains with some β structure. As seen in Figure 3, the laser texturing of the Ti6Al4V surfaces influences the original microstructure up to 1.5–2 μm deep, where the consequences of melting and rapid solidification are observed. In addition, as reported in our previous study, at high cooling rates the Ti6Al4V alloy can be solidified to α’ prime martensite, which is reflected in increased hardness around dimples [28]. The bulk material, however, is not affected by laser texturing at the chosen processing parameters and consists of α + β grains with sizes up to 4 μm.

### 3.1. Tribological Testing

#### 3.1.1. Coefficient of Friction

In the following tests we used three types of synthetic oils with different viscosities (Table 1): T9, VG46, and VG100. Figure 3 presents typical coefficient-of-friction curves for the non-textured and all-laser-textured Ti6Al4V samples (50–100, 50–200, 100–200, and 100–400) under T9, VG46, and VG100 oil lubrication. For the non-textured samples, we can observe relatively stable friction during the sliding experiment in all three oils, regardless of their viscosity. There is, however, a slight difference in the steady-state coefficient of friction. The initial and steady-state COF for T9 is around 0.42, while for VG46 and VG100 they are slightly higher, around 0.44 (Figure 4 and Figure 5). 

Friction for the laser-textured Ti6Al4V in the least-viscous oil, T9 (8.9 mm^2^/s), is characterized by the most pronounced steady-state COF-reduction effect in comparison to the non-textured surface—of around 10%. The best results are obtained for the 50–100 and 100–200 surfaces with the highest surface texture-density, where the average COF were ~0.36 and 0.37, respectively (Figure 4a and Figure 5). Slightly higher COF, yet still significantly lower than the non-textured surface, ~0.39 and 0.38 are obtained for the 50–200 and 100–400 surfaces with the lowest texture density, respectively (Figure 4a and Figure 5). A slight increase in the COF with the reduced TD, which is correlated with the size of the dimples and their center-to-center distances, can be understood by a variation in the micro-hydrodynamic effects of the dimples, leading to increased friction [29,30]. 

The friction behavior of laser-textured samples under more viscous oils, such as VG46 (50.6 mm^2^/s) and VG100 (110 mm^2^/s), leads to increased friction due to a thicker oil film with an increased shearing resistance. Interestingly, however, the friction under VG46 and VG100 is comparable, even though they differ in viscosity by a factor of two. Laser texturing in combination with VG46 and VG100 caused an increase in friction for the 50–100 surface with the highest TD, resulting in the steady-state COF of ~0.45, which is even higher than the friction in these oils for the non-textured surface (Figure 4 and Figure 5). Friction under lubrication with VG46 and VG100 is of the same order for the 50–200 and 100–400 surfaces with the lowest TD, ~0.41 (Figure 4b,c and Figure 5). This COF is also comparable to the COF of the non-textured surface. A friction-reducing effect in comparison to the non-textured Ti6Al4V alloy was only observed for the 100–200 surface, with a comparable steady-state COF of ~0.38 under lubrication with both the VG46 and VG100 oils (Figure 4b,c and Figure 5). A comparable friction behavior for the different textures under VG46/VG100 lubrication suggests that laser texturing overcomes the negative influence of the oil viscosity effect on the friction through the oil retention capacity of the textured surfaces. Overall, regardless of the lubricant type and oil viscosity, the lowest COF was measured for the sample 50–100 with the highest TD.

#### 3.1.2. Wear

When analyzing wear, the wear of the counter-body, i.e., the100Cr6 ball, was below the detection limit due to the more-than-two-times-higher hardness of the 100Cr6 compared to the Ti6Al4V material. Therefore, all the wear was measured on the Ti6Al4V surface, regardless of the texturing pattern and oil lubricant. Figure 6 presents SEM images of the wear scars of the non-textured and all-laser-textured Ti6Al4V samples (50–100, 50–200, 100–200, and 100–400) under T9, VG46, and VG100 oil lubrication.

In terms of wear, we observed a combination of predominantly abrasive and adhesive wear mechanisms for all the Ti6Al4V samples, non-textured and laser-textured. Wear volumes were calculated from the measured widths and depths of the wear scars and are presented in Figure 7. We observed similar wear volumes in the range of 0.0035 mm^3^ for the non-textured Ti6Al4V surface in all three oils. This suggests poor wettability and lubrication conditions for all three oils and critical contact conditions with the boundary lubrication regime, regardless of the oil viscosity. Even the highest viscosity oil does not provide a sufficient oil film to separate the contact surfaces, as also confirmed by intensive abrasive wear marks. Laser texturing, however, seemed to even intensify the abrasive wear component in comparison to the non-textured surface, mostly due to the reduced contact area and the increased contact stresses within the contact [31]. This effect is most pronounced for the highest texture density, sample 50–100, and then diminishes with the reduced texturing density (Figure 7). For the highest texturing density (sample 50–100) the increase in wear volume, as compared to the non-textured contact, is around five-times-under T9 (~0.018 mm^3^) and even ten-times-under VG100 (~0.036 mm^3^) lubrication. This clearly shows the interrelated effect of the oil’s viscosity and the surface-texturing parameters. For very high texturing densities (≥40%), the oil’s viscosity has a negative effect, with all the oil being constrained in the dimples, and the higher the viscosity, the less oil is provided within the actual contact. As the texturing density is reduced, there is a critical combination of oil viscosity and texturing density where a positive oil supply and the micro-hydrodynamic effects of the individual dimples can prevail over the oil-constraining effect. In the case of the low-viscosity oil (T9) with better spreadability, the reduced texturing density results in improved lubrication conditions and less wear, with the best results similar to the non-textured case observed for a texturing density of 15%. However, if the texturing density is too low (≤10%), the oil retention function of the dimples is lost again, leading to increased wear. For medium-viscosity oil (VG 46) and a better oil retention capability, the contact conditions and wear resistance are improved with a reduced texturing density. The best results are observed for a low texturing density of 10%, being the result of the combined effect of a larger contact area, with the dimples acting as oil reservoirs and the oil having sufficient spreadability. However, in the case of the high-viscosity oil (VG100), again the medium texturing densities of about 15% provide the best contact conditions and wear results. Significantly reduced wear (~0.002 mm^3^), even below the non-texturing case, was observed for sample 50–200 under VG100 lubrication, indicating the combined effect of oil reservoirs and an increased oil-film thickness within the dimples due to the high viscosity. Overall, the analysis of the combination of friction and wear suggests that the best candidate is sample 50–200 with a medium TD under low- and high-viscosity-oil lubrication.

## 4. Conclusions

In this research, the tribological behavior of laser-textured Ti6Al4V alloy surfaces was systematically investigated under the influence of three different synthetic oils: T9, VG46, and VG100. Laser texturing was utilized to create dimple patterns on the Ti6Al4V surfaces, with varying dimple diameters and center-to-center distances.

The results show that the laser-textured surfaces exhibited reduced steady-state coefficients of friction (COF) compared to the non-textured surface, particularly under low-viscosity oil (T9). Surfaces with a high texture density (50–100 and 100–200) exhibited the lowest COF, showing a reduction of around 10% compared to the non-textured surfaces. However, under higher-viscosity oils (VG46 and VG100), the laser-textured surfaces exhibited increased friction compared to the non-textured surface. The effect of the oil’s viscosity on the friction was mitigated to some extent by the oil retention capacity of the textured surfaces.

The wear analysis indicated a combination of abrasive and adhesive wear mechanisms on all surfaces, with similar wear volumes observed for the non-textured surfaces under all oil types. Laser texturing was intended to intensify the abrasive wear, especially on surfaces with a high texture density. The interplay between the oil’s viscosity and the surface texture density therefore played a crucial role in the wear resistance. In the case of low- and medium-viscosity oils (T9 and VG46), reduced texturing density results in improved lubrication conditions and less wear. The best results are observed for a low texturing density of 10%, being the result of the combined effects of a larger contact area, dimples acting as oil reservoirs, and oil having sufficient spreadability. On the other hand, in the case of the high-viscosity oil (VG100), medium texturing densities of about 15% provide the best contact conditions and wear results. Finally, significantly reduced wear, even below the non-texturing case, was observed for sample 50–200 under VG100 lubrication, indicating the combined effect of oil reservoirs and increased oil-film thickness within the dimples due to high viscosity.

Overall, the results suggest that laser texturing can effectively reduce the friction under certain lubrication conditions, while the impact of laser texturing on the wear strongly depends on a complex interplay of oil viscosity and surface texture density. These findings offer valuable insights into enhancing the tribological performance of the Ti6Al4V alloy in demanding industrial applications.

## Figures and Tables

**Figure 1 materials-16-06615-f001:**
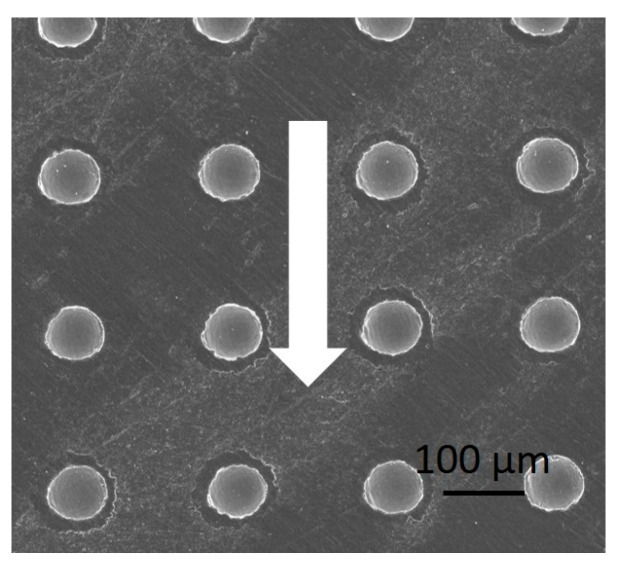
Sliding direction marked by an arrow.

**Figure 2 materials-16-06615-f002:**
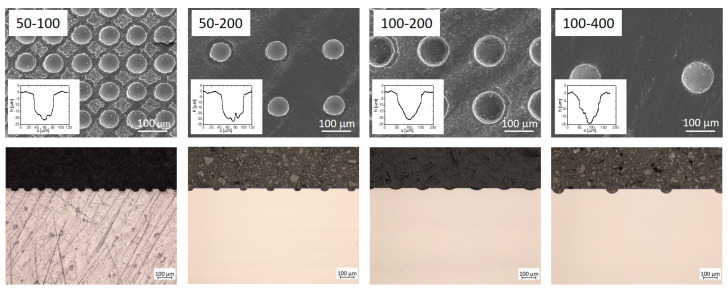
SEM images (15.0 V, X200, WD 36.2) of analyzed dimple-laser-textured Ti6Al4V surfaces, 50–100, 50–200, 100–200, and 100–400; and below, the corresponding light microscopy images. The first number labels the dimple width, while the second number indicates the dimple-to-dimple distance. The inset in each SEM image shows the depth profile of the dimples.

**Figure 3 materials-16-06615-f003:**
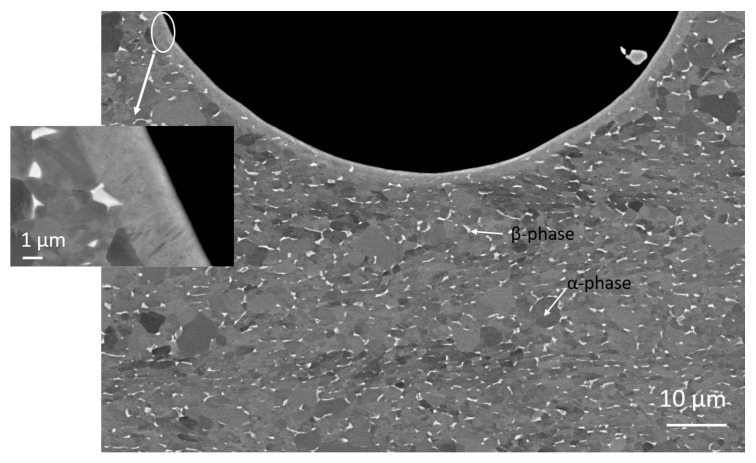
SEM micrograph of a dimple on a 100–400 dimple-textured surface. The inset shows the details of the laser remelted layer.

**Figure 4 materials-16-06615-f004:**
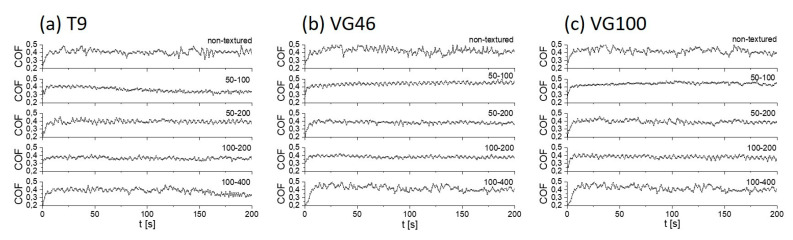
Coefficient-of-friction curves for non-textured and all laser-textured Ti6Al4V samples (50–100, 50–200, 100–200, and 100–400) under T9 (**a**), VG46 (**b**), and VG100 (**c**) oil lubrication.

**Figure 5 materials-16-06615-f005:**
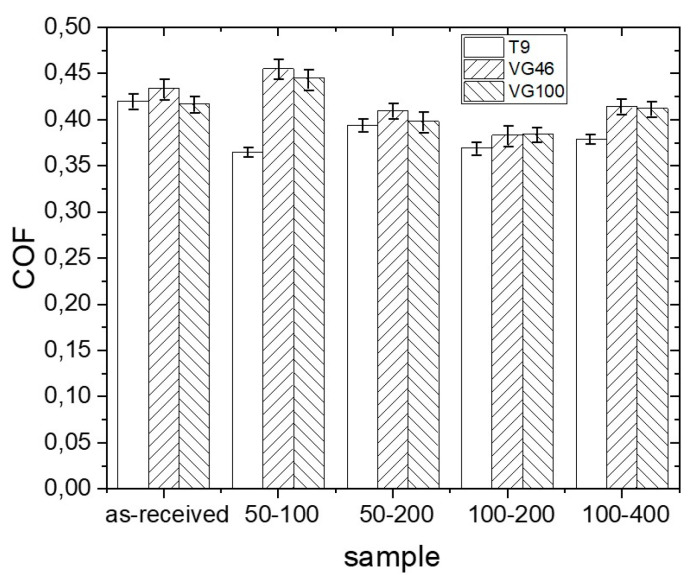
Steady-state coefficient of friction (COF) for non-textured and laser-textured Ti6Al4V surface under T9/VG46/VG100 oil lubrication.

**Figure 6 materials-16-06615-f006:**
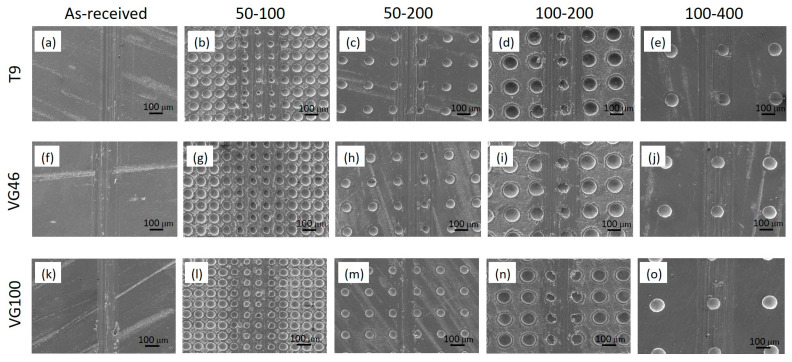
SEM images (15.0V, X100, WD 32.2) of wear scars of non-textured (**a**,**f**,**k**) and laser-textured Ti6Al4V surfaces under T9 (**b**–**e**), VG46 (**g**–**j**) and VG100 (**l**–**o**) oil lubrication.

**Figure 7 materials-16-06615-f007:**
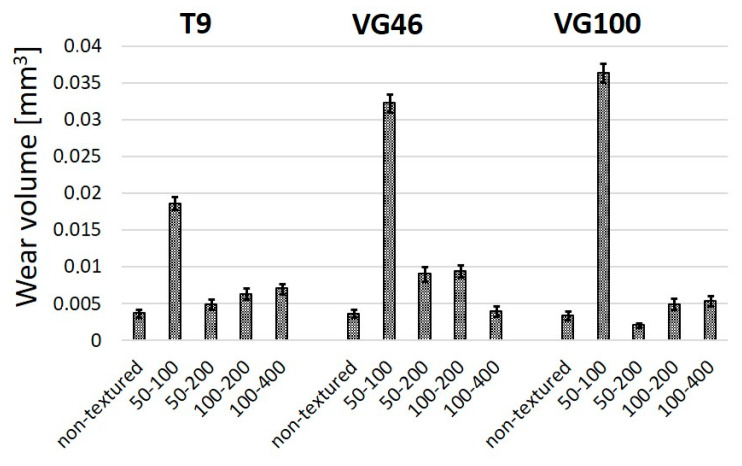
Wear volumes of non-textured and laser-textured Ti6Al4V surfaces under T9/VG46/VG100 oil lubrication.

**Table 1 materials-16-06615-t001:** Oil viscosity as reported in the data sheet by the provider.

Oil	Viscosity at 40° [mm^2^/s]
T9	8.9
VG46	50.6
VG100	110

**Table 2 materials-16-06615-t002:** Parameters of the laser texturing.

Pattern Type	Diameter of Circles [µm]	Centre-to-Centre Distance [µm]	Processing Time [s]
50–100	50	100	2508
50–200	50	200	736
100–200	100	200	467
100–400	100	400	220

**Table 3 materials-16-06615-t003:** Texture density of laser-textured surfaces under investigation.

Sample	TD [%]
50–100	40
50–200	15
100–200	30
100–400	10

## Data Availability

Not applicable.

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
