# Peer review of "Influence of Oil Viscosity on the Tribological Behavior of a Laser-Textured Ti6Al4V Alloy"

_materials, 2023, doi:10.3390/ma16196615_

Round 1
Reviewer 1 Report
This paper presented the tribological behaviour of laser-textured Ti6Al4V alloy surfaces under the influence of three different synthetic oils. Regarding to tribological aspects, this study is an important topic and can be of interest to engineering applications. Although the paper is well organized and is easy to follow the writing, the current manuscript needs improvements or additions. To this reviewer, major revisions and further clarifications are needed before a final decision can be rendered regarding this manuscript. The authors need to properly address the following comments to further improve this manuscript:
1. Sort the keywords alphabetically;
2. To emphasize the study gaps that your research seeks to fill, it is advisable to explain the novelty, and limitations of previous studies in the introduction;
3. The third paragraph - Results and discussion- line 109 – must be renamed because this name is appropriate after performing the experiments;
4. In lines 190-192, explain more clearly how wear volumes were determined;
5. Add more information to the results as follows:
- Based on the results of the experiments, state the optimal texturing option for each type of lubricant;
- Also, based on the results of the experiments, state the optimal texturing option for the analysed alloy (with the lowest friction coefficient and the lowest wear volume).
6. The authors should enrich their references in the revised manuscript from five years ago. Prefer inclusion of literature published by MDPI.
Author Response
This paper presented the tribological behaviour of laser-textured Ti6Al4V alloy surfaces under the influence of three different synthetic oils. Regarding to tribological aspects, this study is an important topic and can be of interest to engineering applications. Although the paper is well organized and is easy to follow the writing, the current manuscript needs improvements or additions. To this reviewer, major revisions and further clarifications are needed before a final decision can be rendered regarding this manuscript. The authors need to properly address the following comments to further improve this manuscript:
- Sort the keywords alphabetically;
Thank you for the comment. Keywords were arranged.
- To emphasize the study gaps that your research seeks to fill, it is advisable to explain the novelty, and limitations of previous studies in the introduction;
Thank you for your valuable comment. The Introduction was improved as suggested. Some more references were added.
- The third paragraph - Results and discussion- line 109 – must be renamed because this name is appropriate after performing the experiments;
Thank you for the comment. As the Journal does not demand that we find it more clear to discuss the results together with the presentation of the results.
- In lines 190-192, explain more clearly how wear volumes were determined;
Thank you for your comment. The wear volumes were calculated from the measured widht and depth of the wear scars. This was added to the text.
- Add more information to the results as follows:
- Based on the results of the experiments, state the optimal texturing option for each type of lubricant;
Thank you. The optimal sample is 50-100 with the lowest COF, regardless of the lubricant type. This was added to the text.
- Also, based on the results of the experiments, state the optimal texturing option for the analysed alloy (with the lowest friction coefficient and the lowest wear volume).
Thank you for the comment. Although there are too many factors influencing the choice of the best candidate in terms of friction and wear, we propose medium TD under lubrication with high viscosity oil. This was added to the text.
- The authors should enrich their references in the revised manuscript from five years ago. Prefer inclusion of literature published by MDPI.
Thank you for your valuable comment. References were added to the manuscript.

Reviewer 2 Report
The paper deals with the influence of oil viscosity on tribological behavior of laser-textured Ti6Al4V alloy, however, is necessary to clarify some processes used to perform this analysis. Also, some factual clarifications will be helpful as highlighted in the comments below.
-Include all the norms used for the tests.
-State clearly the novelty in the introduction.
-Any change on the surface roughness has tribological effects, what is the original roughness previous to the laser texturing process?
-Is possible to define the relationship of surface texturing with a capillary effect on the lubricant used?
-Page 2, line 85. “The laser-texturing was done in air at ambient temperature”. What is the temperature effect on the wearing resistance?
Author Response
The paper deals with the influence of oil viscosity on tribological behavior of laser-textured Ti6Al4V alloy, however, is necessary to clarify some processes used to perform this analysis. Also, some factual clarifications will be helpful as highlighted in the comments below.
-Include all the norms used for the tests.
Thank you for the comment All the experiments were performed as stated in experimental section. The results of COF and wear volumes presented in Figs. 4 and 6 are based on 3 independent measurements and standard deviation is included in the graphs.
-State clearly the novelty in the introduction.
Thank you for the comment. The Introduction was improved as suggested.
-Any change on the surface roughness has tribological effects, what is the original roughness previous to the laser texturing process?
Thank you for your comment. The average surface roughness of the original Ti6Al4V surface is stated in the Experimental section and is Sa = 0.17 µm.
-Is possible to define the relationship of surface texturing with a capillary effect on the lubricant used?
Good point. However, the focus of the current research was based on friction and wear and we didn’t study the effect of capillary force.
-Page 2, line 85. “The laser-texturing was done in air at ambient temperature”. What is the temperature effect on the wearing resistance?
Thank you for the comment. The present study was performed at room temperature. It can be expected that elevated temperatures would have influence on wear resistance, which is the subject of our further work.

Round 2
Reviewer 1 Report
The answers are appropriate
Author Response
Thank you for the reply.